# Protective Effects and Mechanism of Polysaccharides from Edible Medicinal Plants in Alcoholic Liver Injury: A Review

**DOI:** 10.3390/ijms242216530

**Published:** 2023-11-20

**Authors:** Zhuo-Wen Su, Ting-Yu Yan, Jing Feng, Meng-Yuan Zhang, Lei Han, Hua-Feng Zhang, Ying Xiao

**Affiliations:** 1National Engineering Laboratory for Resource Development of Endangered Crude Drugs in Northwest China, College of Food Engineering and Nutritional Science, International Joint Research Center of Shaanxi Province for Food and Health Sciences, Shaanxi Normal University, Xi’an 710119, China; suzhuowen87811@163.com (Z.-W.S.);; 2Academician and Expert Workstations in Puer City of Yunnan Province, Puer 665600, China; 3Faculty of Medicine, Macau University of Science and Technology, Macau SAR, China

**Keywords:** polysaccharides, edible medicinal plants, alcoholic liver injury, hepatoprotective effects, alcoholism, mechanism of action

## Abstract

Alcohol use accounts for a large variety of diseases, among which alcoholic liver injury (ALI) poses a serious threat to human health. In order to overcome the limitations of chemotherapeutic agents, some natural constituents, especially polysaccharides from edible medicinal plants (PEMPs), have been applied for the prevention and treatment of ALI. In this review, the protective effects of PEMPs on acute, subacute, subchronic, and chronic ALI are summarized. The pathogenesis of alcoholic liver injury is analyzed. The structure–activity relationship (SAR) and safety of PEMPs are discussed. In addition, the mechanism underlying the hepatoprotective activity of polysaccharides from edible medicinal plants is explored. PEMPs with hepatoprotective activities mainly belong to the families Orchidaceae, Solanaceae, and Liliaceae. The possible mechanisms of PEMPs include activating enzymes related to alcohol metabolism, attenuating damage from oxidative stress, regulating cytokines, inhibiting the apoptosis of hepatocytes, improving mitochondrial function, and regulating the gut microbiota. Strategies for further research into the practical application of PEMPs for ALI are proposed. Future studies on the mechanism of action of PEMPs will need to focus more on the utilization of multi-omics approaches, such as proteomics, epigenomics, and lipidomics.

## 1. Introduction

Alcohol is a commonly misused substance that has substantial negative impacts on human health [1,2]. Alcohol consumption is considered a leading risk factor for mortality, and it accounted for approximately 1.8 million deaths in 2020 [3]. Unfortunately, the global adult per capita consumption of alcohol has continuously increased in recent years, and it is forecasted to reach 7.6 L by 2030 [4]. In the human body, the liver is the primary organ for alcohol catabolism and detoxification, and long-term or excessive alcohol exposure is likely to cause alcoholic liver injury (ALI), which has become one of the most prevalent diseases worldwide [5]. Frequently, ALI initially manifests as fatty liver disease, which further develops into alcoholic steato-hepatitis, alcoholic hepatic fibrosis, and alcoholic cirrhosis. In particular, severe alcoholism may result in large-scale hepatocellular necrosis and even liver cancer [6,7]. Therefore, it is urgent to seek medicines or nutraceuticals to prevent and treat ALI and to elucidate their mechanisms of action.

Edible medicinal plants have been used in the food and drug industries in China for over 2000 years due to their low toxicity and high efficiency [8,9]. To date, a vast array of compounds in edible medicinal plants have been found to possess protective abilities against ALI. In our laboratory, the protective effects of flavonoids, like epimedin C isolated from *Epimedium sagittatum*, on acute alcoholism were systematically investigated [10]. It has been observed that epimedin C significantly shortens the lethargy time of mice suffering from acute alcoholism; reduces the levels of aspartate transaminase (AST), alanine transaminase (ALT), total bilirubin (TBIL), total cholesterol (TC), and triglyceride (TG) in serum; increases the activity of superoxide dismutase (SOD) and glutathione peroxidase (GSH-Px) in the liver; improves the content of reduced glutathione (GSH) in the liver; and decreases the level of malondialdehyde (MDA) in the liver. Moreover, epimedin C remarkably regulates protein expression in mice, repairs the damaged nervous system, and relieves disorders of carbohydrate and lipid metabolism [10]. To sum up, epimedin C possesses great potential for preventing and treating ALI, partially due to its antioxidant ability and regulatory effects on protein expression. Likewise, a vast array of polysaccharides from edible medicinal plants (PEMPs) exhibit obvious hepatoprotective activities [11], some of which are increasingly recognized as potential sources of phytomedicines and/or functional foods for the prevention and treatment of alcoholic liver injury. The bioavailability of plant polysaccharides mainly depends on three pathways: (1) direct absorption; (2) absorption through intestinal microflora; and (3) absorption through Peyer’s patches [12]. It is of theoretical and practical significance to study the effects of PEMPs against ALI.

This review summarizes the protective effects of polysaccharides from edible medicinal plants on alcoholic liver injury and explore their mechanism of action. In particular, the structure–activity relationship (SAR) of PEMPs will be analyzed. In addition, further strategies for the scientific exploration and comprehensive utilization of PEMPs will be proposed. Taking into account their similarity to plants, some edible medicinal fungi will also be discussed in terms of polysaccharides with hepatoprotective activities.

## 2. Protective Effects of Polysaccharides from Edible Medicinal Plants on Alcoholic Liver Injury

### 2.1. Protective Activities of PEMPs against Acute Alcoholic Liver Injury

Usually, acute alcoholic liver injury is a toxic effect on the liver caused by a single exposure or multiple exposures to alcohol within 24 h. A great diversity of PEMPs display in vitro or in vivo hepatoprotective activities against acute ALI (Table 1). In general, edible medicinal plants containing these polysaccharides mainly belong to the families Moraceae, Orchidaceae, Asphodelaceae, Apiaceae, Menispermaceae, and Schisandraceae (Table 1). In ALI mice, treatment with 50 mg/kg *Morus alba* fruit (Mori Fructus) polysaccharides (MFPs) effectively reduced the activities of ALT and AST, decreased the contents of TG and MDA, elevated the activity of SOD, and protected liver cells and tissues [13]. Interestingly, sulfated polysaccharides (S-MFPA1, S-MFPA2, S-MFPB1, and S-MFPB2) derived from MFPs were also observed to reduce the levels of serum AST, ALT, and TG; increase the activities of SOD and GSH-Px in the liver; inhibit the production of MDA; and activate alcohol dehydrogenase (ADH) [14]. Polysaccharides from *Angelica sinensis* notably prevented hepatic steatosis and fat accumulation, which were related to the up-regulation of hepatic hypoxia-inducible factor 1-alpha (HIF-1α) [15]. *Ganoderma lucidum* polysaccharides were found to inhibit the expression of the nucleotide-binding oligomerization domain-like receptor protein 3 (NLRP3) inflammasome and alleviate liver fat deposition in ALI mice [16]. In addition, *Ganoderma lucidum* polysaccharides drastically affected the metabolism of choline and glycerophospholipids and regulated adenosine triphosphate (ATP)-binding cassette (ABC) transporters in mice [17].

### 2.2. Protective Activities of PEMPs against Subacute Alcoholic Liver Injury

Generally, subacute alcoholic liver injury refers to a liver injury caused by repeated exposure to exogenous alcohol for 14–28 consecutive days. There have been several reports on PEMPs with hepatoprotective effects on subacute ALI so far (Table 2). These polysaccharides were frequently prepared from edible medicinal plants of the families Orchidaceae, Phallaceae, and Rosaceae (Table 2). Polysaccharides from *Poria cocos* exhibited the following protective activities against subacute ALI: (1) inhibiting the activities of SOD, ALT, and AST; (2) reducing the content of MDA; (3) decreasing the levels of interleukin 6 (IL-6) and tumor necrosis factor-α (TNF-α); (4) regulating the expression of cytochrome P4502E1 (CYP2E1); and (5) suppressing the Toll-like receptor 4/nuclear factor-kappa B (TLR4/NF-κB) inflammatory signaling pathway [27]. Acidic polysaccharides from *Panax notoginseng* could activate nuclear factor erythroid 2-related factor 2 (Nrf2) in ALI mice, strengthen the ADH pathway, and inhibit the catalase (CAT) pathway of alcohol metabolism [28]. The hepatoprotective capacities of polysaccharides from six *Dendrobium* species (polysaccharides from *Dendrobium huoshanense* (DHP), *Dendrobium officinale* (DOP), *Dendrobium fimbriatum* (DFP), *Dendrobium chrysotoxum* (DCP), *Dendrobium nobile* (DNP), and *Dendrobium moniliforme* (DMP)) were comparatively investigated. Noticeably, the protective activities of DHP and DOP were higher than those of DFP, DCP, DNP, and DMP [29]. DHP could suppress the activities of ALT, AST, and alkaline phosphatase (ALP); decrease the levels of high-density lipoprotein cholesterol (HDL-C), low-density lipoprotein cholesterol (LDL-C), TC, and TG in serum; improve the activities of ADH, acetaldehyde dehydrogenase (ALDH), SOD, and GSH-Px in the liver; and inhibit GSH depletion and MDA accumulation in the liver [30]. Furthermore, DHP obviously ameliorated the abnormalities of phosphocholine and L-proline metabolism and regulated the expression of the cystathionine beta-synthase (Cbs) and D-Lactate dehydrogenase (Ldhd) genes [31,32].

### 2.3. Protective Activities of PEMPs against Subchronic Alcoholic Liver Injury

In subchronic ALI tests, experimental animals (mainly rodents) are usually subjected to continuous alcohol exposure for 1–6 months (no more than 10% of their life cycle). The majority of PEMPs with protective abilities against subchronic ALI belong to the families Solanaceae, Rhamnaceae, Liliaceae, Asteraceae, and Acanthaceae (Table 3). *Ziziphus jujube* (jujube) polysaccharides could elevate the activities of SOD, CAT, and GSH-Px in ALI rats and decrease the contents of MDA in the liver, heart, spleen, kidneys, and lungs [38]. Similarly, *Allium sativum* (garlic) polysaccharides could increase the activities of SOD and GSH-Px and reduce the content of MDA [39]. Meanwhile, garlic polysaccharides dramatically modulated the protein expressions of transforming growth factor-beta 1 (TGF-b1) and TNF-α [39]. Notably, gastric infusions of *Lycium barbarum* polysaccharides (LBPs) for 30 consecutive days reduced the activities of serum ALT and AST and prevented fatty liver induced by alcohol [40]. Additionally, LBPs could attenuate alcohol-induced oxidative stress and cellular apoptosis, both of which were possibly attributed to the inhibition of thioredoxin-interacting protein (TXNIP) [41].

### 2.4. Protective Activities of PEMPs against Chronic Alcoholic Liver Injury

Long-term exposure to exogenous alcohol may lead to chronic alcoholic liver injury. PEMPs with protective effects on chronic ALI were mainly extracted from edible medicinal plants of the families Solanaceae, Liliaceae, Asphodelaceae, Lauraceae, and Lamiaceae (Table 4). Polysaccharides from *Aloe barbadensis* (AVGPs) effectively attenuated the activities of serum aminotransferases in ALI mice, reduced the levels of hepatic TG and TNF-α, up-regulated lipolytic proteins such as adenosine monophosphate-activated protein kinase-α2 (AMPK-α2) and proliferator-activated receptor-α (PPAR-α), and down-regulated TLR4 and myeloid differentiation primary response 88 (MyD88) [44]. Polysaccharides from *Ganoderma lucidum* exerted a protective ability against chronic ALI in mice by improving lipid metabolism and reducing inflammatory factors in hepatic cells [45]. Intriguingly, the combination of LBPs and ZnSO_4_ had a synergistic effect on chronic ALI in rats, which could improve lipid metabolism, inhibit oxidative stress, control inflammatory responses, and regulate the activities of enzymes related to alcohol metabolism [46].

## 3. Mechanism of PEMPs Underlying Protective Effects on ALI

### 3.1. Pathogenesis of Alcoholic Liver Injury

The pathogenesis of alcoholic liver injury is quite complex and may include (1) the detrimental effects of alcohol metabolites; (2) the influences of inflammatory mediators and/or cytokines; (3) the impact of endotoxin; (4) disorders of lipid metabolism; (5) the immune response; (6) oxidative stress; and (7) apoptosis [50,51,52,53,54,55,56,57,58,59,60,61]. Among alcohol metabolites, acetaldehyde is regarded as one of the most deleterious substances as it interferes with several functions of hepatocytes (Figure 1). Acetaldehyde may increase the level of transforming growth factor beta (TGF-β) and stimulate hematopoietic stem cells (HSCs), which results in liver fibrosis [55]. In particular, acetaldehyde–protein adducts (APAs), which are formed by the covalent conjugation of acetaldehyde and proteins, may decrease protease activity, consume glutathione, and eventually induce lipid peroxidation and hepatocyte apoptosis [62]. Generally, oxidative stress accelerates ALI by altering biofilm functions; inhibiting enzyme activity; and activating NF-κB, cyclooxygenase (COX-2), and TNF-α [63]. Alcohol-induced mitochondrial DNA damage may affect electron-transfer reactions, attenuate the β-oxidation of fatty acids, reduce ATP production, and enhance the expression of apoptotic signaling molecules such as caspases [64]. Furthermore, excessive alcohol consumption may cause abnormal transcription of hepcidin, promote the expression of iron transporters, increase the intestinal absorption of iron, and then lead to iron overload in the liver [65].

### 3.2. Mechanism of Action of PEMPs

#### 3.2.1. Activation of Enzymes Related to Alcohol Metabolism

The gastric mucosa serves as the first site of alcohol catabolism. Alcohol dehydrogenase present in the gastric mucosa facilitates the first-pass metabolism (FPM) of alcohol in the stomach, which reduces the bioavailability of alcohol and alleviates the toxic effects of alcohol on organs such as the liver and brain [66]. Thus, increasing ADH activity and enhancing FPM in the stomach are beneficial for the liver. After passing through the gastrointestinal system, alcohol is mainly metabolized in the liver. Usually, the metabolic pathways of alcohol include (1) the microsomal ethanol-oxidizing system (MEOS) in the endoplasmic reticulum; (2) the ADH system in liver cytoplasm; and (3) the CAT system [67]. All three metabolic pathways produce harmful acetaldehyde (Figure 2). ALDH catalyzes the oxidation of acetaldehyde into acetic acid, which enters the tricarboxylic acid cycle and is subsequently oxidized to CO_2_ and H_2_O [50]. Excessive alcohol consumption may decrease ADH activity, increase the CYP2E1 activity in microsomes, and cause MEOS hyperactivity, which generates large amounts of acetaldehyde and reactive oxygen species (ROS) [68]. Presently, there have been numerous reports dealing with the effects of PEMPs on ADH and ALDH activation. In our laboratory, a novel fraction of polysaccharides (EP80) was prepared from *Epimedium sagittatum*, which obviously activated ADH and ALDH [69]. Polysaccharides from *Dendrobium huoshanense* effectively increased the activities of ADH and ALDH in the liver and mitigated hepatocyte damage in mice [30]. Polysaccharides from *Ophiopogon japonicus* notably increased the activities of ADH and ALDH in the liver and decreased the concentrations of alcohol and acetaldehyde in serum [70]. The activating effects of three polysaccharides (MFP-1, MFP-2, and MFP-3) from *Morus alba* fruits on ADH were stronger than that of tiopronin [71]. It is noticeable that alcohol metabolism involves a series of enzymatic reactions (Figure 1). When compared to ADH and ALDH, investigations into other key enzymes or proteins (e.g., CAT and CYP2E1) are lacking. Therefore, future research into the activation of enzymes related to alcohol metabolism will need to focus more on other enzymes or proteins aside from ADH and ALDH.

#### 3.2.2. Attenuation of Damage from Oxidative Stress Caused by Alcohol Intake

In the human body, the presence of a small number of free radicals generally maintains the redox balance. However, the balance may be disturbed by the overproduction of ROS resulting from excessive alcohol intake. Subsequently, oxidative stress may emerge as an important contributor to alcoholic liver injury [72]. ROS may damage DNA, proteins, and lipids in hepatocytes; exacerbate mitochondrial dysfunction; promote apoptosis; and trigger liver fibrosis [73]. Since both the in vivo antioxidant defense system (e.g., SOD and GSH-Px) and in vitro antioxidants (e.g., ascorbic acid) could scavenge ROS or free radicals (Figure 2), the effects of PEMPs on the antioxidant defense system in ALI animal models were extensively investigated [74]. Polysaccharides from *Rosa rugosa* cure ALI by enhancing the activities of SOD and GSH-Px, increasing the level of GSH, and decreasing the contents of nitric oxide (NO) and MDA [35]. Likewise, two fractions (MFPA1 and MFPB1) derived from *Morus alba* fruit polysaccharides could suppress oxidative stress in ALI mice [14]. A great number of studies indicated that ALI was intimately linked with oxidative stress [75]. And many PEMPs exerted protective effects on ALI by enhancing the activities of antioxidant enzymes such as SOD and GSH-Px, reducing the content of MDA, and increasing the content of GSH [76].

#### 3.2.3. Regulation of Cytokines

Cytokines are a family of small, secreted proteins with a wide range of biological activities. They are synthesized by immune or non-immune cells. They can regulate innate and adaptive immunity, modulate cell growth, and repair damaged tissues [77,78,79]. In a healthy human, anti-inflammatory and pro-inflammatory factors are in a state of dynamic balance. Once the balance is broken, dysfunction may occur [80]. Alcohol-induced liver inflammation is generally initiated by the activation of Kupffer cells in the presence of pro-inflammatory factors generated during alcohol metabolism, leading to a cascade of inflammatory cytokine production (Figure 1 and Figure 2). In addition, alcohol may induce the diffusion of endotoxins and thereby promote the expression of inflammatory mediators such as interleukin-1 (IL-1), IL-6, and TNF-α [81]. Since the structure and function of the liver may be irreversibly destroyed under the impact of inflammatory factors, anti-inflammation has received extensive attention as an important target of liver protection. Polysaccharides from *Cinnamomum camphora* have been found to mitigate liver inflammation by suppressing the generation of pro-inflammatory cytokines such as interleukin-1β (IL-1β), IL-6, and TNF-α [42]. Similarly, *Zingiber officinale* (ginger) polysaccharides greatly reduced the expression of pro-inflammatory factors, such as IL-6, IL-1β, and TNF-α, in the serum of mice, decreased the contents of endotoxins, and thereby alleviated the inflammatory response [82]. Polysaccharides from *Dictyophora rubrovalvata* clearly ameliorated alcoholic liver injury by regulating the TLR4/nuclear factor-kappa B subunit p65 (NF-кB p65) signaling pathway [34].

#### 3.2.4. Inhibition of Apoptosis of Hepatocytes

Apoptosis, sometimes called programmed cell death, is often considered a self-destruction mechanism to protect cells from external disturbances [72]. Alcohol intake might induce endoplasmic reticulum (ER) stress (Figure 2) and stimulate the link between the ER adaptor and interferon regulatory factor 3 (IRF3), which contribute to the apoptosis of hepatocytes in the presence of B-cell lymphoma 2 (Bcl2)-associated X protein (Bax) [72]. Polysaccharides from *Sinomenium acutum* significantly ameliorated alcoholic liver injury by blocking the apoptosis of liver cells [21]. Polysaccharides from *Cordyceps sinensis* significantly decreased the mRNA levels of caspase-3, caspase-8, caspase-9, and apoptosis antigen-1 (APO-1) and under-regulated the protein expression of cleaved caspase-3 and cleaved caspase-8, which effectively inhibited the apoptosis of human hepatocytes [83]. The combination of polysaccharides from *Hedysarum polybotrys* and salidroside from *Rhodiola rosea* could suppress early apoptosis by elevating the expression of protein kinase C-beta (PKC*β*), Bcl-2, and phospho-extracellular signal-regulated kinases 1 and 2 (phospho-ERK1/2) and decreasing the levels of Bax, cytochrome C (Cyt-C), and cleaved caspase-3 [84].

#### 3.2.5. Improvement of Mitochondrial Function

Typically, mitochondrial abnormalities occur in the early stage of alcoholic liver injury and continuously produce detrimental effects in hepatocytes (Figure 2). Changes in mitochondrial structure and function mainly include the emergence of oxidative stress, alterations to permeability transition pores, abnormal calcium regulation, and damage to mitochondrial proteins [85]. When the liver is exposed to ethanol for a long time, mitochondria may generate a large amount of reactive nitrogen (RNS) and ROS, which may induce lipid peroxidation. Products of lipid peroxidation may inactivate a component of the respiratory chain by destroying mitochondrial DNA (mtDNA), induce mitochondrial permeability transition, cause mitochondrial swelling and rupture, and eventually lead to hepatocyte death [86]. *Lycium barbarum* polysaccharides may cure acute liver injury by augmenting mitochondrial respiration, reactivating respiratory chain complexes I-V, raising tissue ATP levels, alleviating oxidative stress, and regulating metabolic pathways [87]. The mitochondrial membrane potential (MMP) and ATP are pivotal biomarkers to assess mitochondrial function, and the MMP modulates the activities of some enzymes that attenuate liver mitochondrial damage. *Chrysanthemum indicum* polysaccharides and their phosphorylated derivates treated duck hepatitis by increasing ATP production and stabilizing the mitochondrial membrane potential, thus protecting mitochondrial function [88]. Similarly, polysaccharides from *Cyclocarya paliurus* and *Polygonatum sibiricum* improved mitochondrial function and mitigated oxidative damage in human cells such as the L02 hepatic cell line [89,90].

#### 3.2.6. Regulation of Gut Microbiota

In many cases, the gut microbiota plays important roles in human diseases, such as alcoholic liver injury and hepatoma [91,92]. There is usually a bidirectional communication system between the gut microbiota and host cells [93], which is essential to maintain biological functions such as immunomodulation, metabolic reactions, and pathogen elimination [94]. Excessive alcohol consumption may increase intestinal permeability and disrupt intestinal microecology [95]. It is evident that some PEMPs have the ability to regulate intestinal flora and alleviate alcohol-induced intestinal microecological disorders through the gut–liver axis [96]. For this reason, much attention has been focused on PEMPs for the prevention and treatment of ALI by regulating the intestinal microbiome and promoting beneficial microbe–host interactions (Figure 2). For example, *Echinacea purpurea* polysaccharides (EPPs) have been demonstrated to alleviate inflammation and oxidative stress in the liver caused by excessive alcohol intake through the gut–liver interaction, which increases the abundance of *Muribaculaceae*, *Lactobacillus,* and *Bacteroides* while decreasing the abundance of *Escherichia-Shigella*, *Enterococcus*, and *Akkermansia*. That is to say, EPPs ameliorate alcohol-induced disturbances in the gut microbiota and maintain intestinal barrier integrity [97]. Selenylated polysaccharides from *Auricularia auricula* increase the abundance of *Bacteroides*, *Desulfovibrio*, *Pseudomonas*, *Eisenberger*, *Flavobacterium*, *Rikenella*, *Corynebacterium*, *Clostridium*-*ASF*356, *Deferribacterium,* and *Ruminococcus,* whereas the abundance of *Escherichia-Shigella* is down-regulated in the gut microbiota of ALI mice [98]. These findings are consistent with previous results indicating that alcohol consumption may increase *Escherichia-Shigella* abundance in the mouse intestinal tract [99]. *Escherichia-Shigella* is capable of producing endogenous ethanol, which enters liver tissue via the circulation and, subsequently, induces hepatic damage [100]. On the contrary, *Bacteroides*, one of dominant beneficial bacteria, may alleviate liver damage caused by alcohol as it provides nutrients like vitamins to the host and other intestinal microbial residents [101].

## 4. Perspectives

### 4.1. Structure–Activity Relationship (SAR) of PEMPs

It is known that chemical structure is the basis of the biological activities of polysaccharides. Generally, the SAR refers to the relationship between the chemical structure (including the conformation) of polysaccharides and their bioactivities, which has drawn great attention in the fields of glycobiology and glycochemistry. Usually, the hepatoprotective effects of PEMPs are closely related to their chemical structure, especially the molecular weight, monosaccharide composition, glycosidic bond type and position, branching degree, and chain conformation [102,103]. At present, there are few studies on the roles of each functional group in PEMPs’ activities, which hampers the elucidation of the SAR and the development of innovative ALI drugs or food additives. In many cases, PEMPs’ activities are not strong enough to treat human diseases. In other words, although many PEMPs have been found to have hepatoprotective activities, their activities are weak when compared to commercial hepatoprotective medicines such as silymarin capsules. Therefore, the structural modification of PEMPs based upon the SAR is an important direction in the research and development of PEMPs. The strategies for chemical modification include changing the functional groups, regulating the molecular weight, altering the hydrophobicity, and grafting chemical groups, which are mainly realized by means of sulfation, phosphorylation, selenization, acetylation, carboxymethylation, esterification, and oxidation [74,104]. For example, polysaccharides isolated from *Trichosanthes kirilowii* peels were changed into phosphorylated polysaccharides, which exhibited a stronger anti-aging effect on D-galactose-induced aging mice [105]. Sulfated polysaccharides of *Cucurbita moschata* (pumpkin) modified via a chlorosulfonic acid–pyridine method possess a higher scavenging ability against superoxide anions than non-sulfated polysaccharides [106]. To date, the SAR of PEMPs has been less completely characterized partly due to the complexity of the polysaccharide structure. Therefore, further investigations are warranted with the assistance of innovative science and emerging technologies (e.g., artificial intelligence algorithms).

### 4.2. Clarification of Mechanism of Action Using Multi-Omics

Omics techniques have ushered in a research field of complex biochemical interactions and the mechanisms underlying the macro-regulation of dynamic network systems in organisms with the development of computational biology and systems biology. Omics includes genomics, epigenomics, transcriptomics, proteomics, metabolomics, lipidomics, and ionomics [107], which may provide clues regarding the mechanism of action of PEMPs. During alcohol exposure, retinoic acid-induced SH-SY5Y cells were treated with *Epimedium sagittatum* extract, and label-free proteomics was employed to unveil its mechanism of action in our laboratory. It was found that G protein plays critical roles in the alcoholism pathway [108]. Most recently, multi-omics has frequently been used to overcome the drawbacks of single omics [109]. For example, a cross-omics analysis based on transcriptomics and proteomics techniques was performed to elucidate the mechanisms underlying the synergistic effect of aflatoxin M1 and ochratoxin A on Caco-2 cells [110]. In some cases, proteomics may provide information that cannot be obtained using genomics. For instance, though genome-wide data are capable of identifying disease genes, they are insufficient to reveal the correlation between genetic information and disease risk as well as the functional differences of genes among individuals, which may be solved via proteomics analysis [111]. An integrated platform is the core of a multi-omics assay, and functional annotation is the main strategy for a comprehensive analysis. Data on specific biological functions (e.g., fatty acid metabolism) are normalized at different omics levels such as genomics, transcriptomics, proteomics, and metabolomics, and their correlations are analyzed. An integrative network approach was performed to identify common biomarkers for four cancers based on RNA sequencing and DNA methylation data. And the results revealed that some genes (e.g., G protein subunit gamma 11 (GNG11), chromobox 2 (CBX2), and cyclin-dependent kinase 3 (CDKN3)) presented similar survival and prognostic behaviors, which might provide new targets for cancers [112]. In the future, further investigations will be required to clarify the mechanism of action of PEMPs against ALI using multi-omics techniques.

### 4.3. Systematic Investigation of Safety of PEMPs

PEMPs with hepatoprotective activities show great potential to become functional foods and drugs for the prevention and treatment of ALI. In many countries such as China, PEMPs are considered new food materials that need to be subjected to systematic evaluations of toxicity in order to guarantee their safety [113,114]. According to the National Health Commission of China, some species (e.g., *Epimedium sagittatum*) belong to edible medicinal plants, for which the isolation of polysaccharides is officially permitted, whereas for some species (e.g., *Dysosma versipellis*), the use of plant materials for the isolation of PEMPs is forbidden due to their toxicity [8,10]. In many cases, edible medicinal plants may suffer from contamination of microorganisms (e.g., aflatoxin) during the storage period [115], which may affect the safety of PEMPs. Meanwhile, PEMPs may be contaminated by exogenous hazardous substances. For example, the Sevag method, based on chloroform, was usually used to eliminate protein impurities during the extraction of PEMPs, and there was still some residual chloroform in polysaccharide samples. To provide further information on the safety of PEMPs, the following in vitro and in vivo toxicological analyses should be performed: (1) the determination of heavy metals; (2) the evaluation of neurobehavioral toxicity, endocrine toxicity, reproductive and developmental toxicity, individual organ toxicity, immunotoxicity, genotoxicity, and carcinogenesis; and (3) toxicokinetic analyses (including absorption, distribution, metabolism, and excretion).

## 5. Conclusions

At present, the most common therapy to cure ALI uses chemosynthetic drugs, which have some limitations. Alternatively, edible medicinal plants, together with fungi, may become an important source of bioactive polysaccharides with relatively strong effects against ALI. A great number of polysaccharides from edible medicinal plants exhibited evident protective effects on acute, subacute, subchronic, and chronic alcoholic liver injuries. The edible medicinal plants used to prepare polysaccharides with hepatoprotective activities mainly belong to the families Orchidaceae, Solanaceae, and Liliaceae. The possible mechanisms by which PEMPs exert their hepatoprotective capacities are as follows: (1) the activation of enzymes related to alcohol metabolism; (2) the attenuation of damage resulting from oxidative stress; (3) the regulation of cytokines; (4) the inhibition of apoptosis of hepatocytes; (5) the improvement of mitochondrial function; and (6) the regulation of the gut microbiota. Future research into the hepatoprotective effects and mechanism of action of PEMPs will need to focus more on safety and the structure–activity relationship, as well as multi-omics approaches such as proteomics, epigenomics, and lipidomics. Additionally, in order to accelerate the clinical application of PEMPs for the prevention and treatment of ALI, it is necessary to improve the bioavailability of PEMPs.

## Figures and Tables

**Figure 1 ijms-24-16530-f001:**
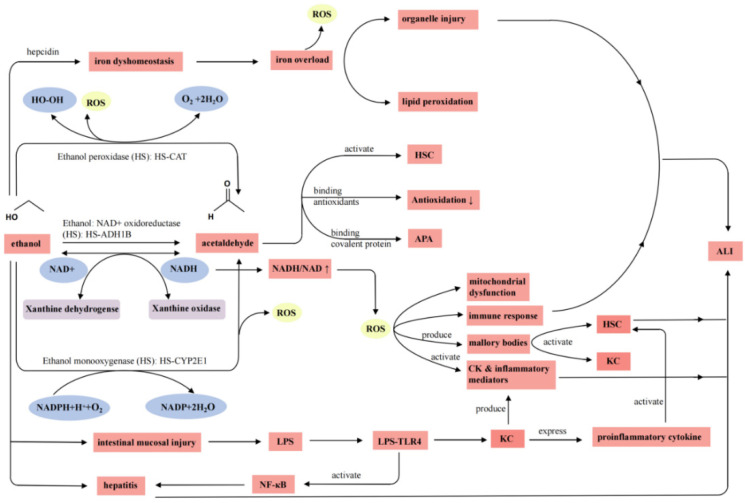
Pathogenesis of alcoholic liver injury. Note: ROS, reactive oxygen species; NAD+, nicotinamide adenine dinucleotide; NADH, reduced nicotinamide adenine dinucleotide; NADP, nicotinamide adenine dinucleotide phosphate; NADPH, reduced nicotinamide adenine dinucleotide phosphate; HSC, hepatic stellate cell; LPS, lipopolysaccharide; APA, acetaldehyde–protein adduct; CK, cytokine; KC, Kupffer cell; TLR4, Toll-like receptor 4; NF-κB, nuclear factor-kappa B; ALI, alcoholic liver injury.

**Figure 2 ijms-24-16530-f002:**
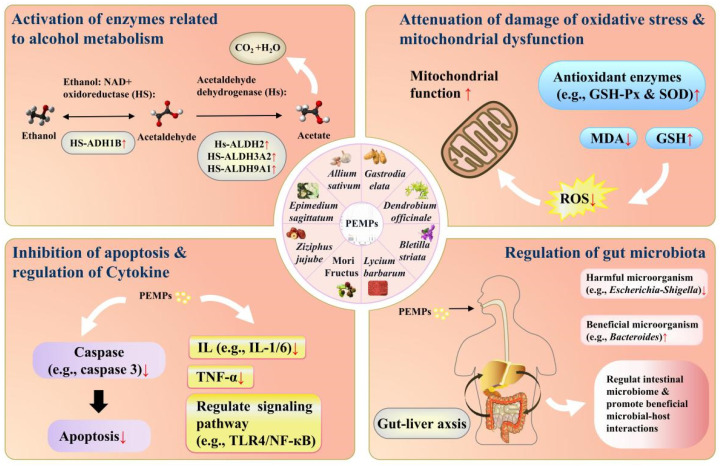
Mechanism of action of PEMPs. Note: PEMPs, polysaccharides from edible medicinal plants; NAD+, nicotinamide adenine dinucleotide; ADH, alcohol dehydrogenase; ALDH, acetaldehyde dehydrogenase; GSH-Px, glutathione peroxidase; MDA, malondialdehyde; GSH, reduced glutathione; ROS, reactive oxygen species; IL, interleukin; TNF-α, tumor necrosis factor-α; TLR4/NF-κB, Toll-like receptor 4/nuclear factor-kappa B.

**Table 1 ijms-24-16530-t001:** PEMPs with hepatoprotective activities against acute alcoholic liver injury.

Family	Source of Polysaccharides (Species)	Names of Polysaccharides	Molecular Weight (kDa)	Monosaccharide Composition	Dose (mg/kg)	Bioactivity	Reference
Moraceae	*Morus alba*	MFPA1MFPB1	/	/	50 50	Normalized lipid metabolism; reduced AST, ALT, TG, and MDA levels; increased GSH and SOD levels; and repaired liver injury in mice.	[13]
Moraceae	*Morus alba*	MFPA1	177	Man/Rha/Glc/Xyl = 6.80:7.40:36.60:46.20	50	MFPA1 had the strongest ability to activate ADH in vitro. MFPA1, MFPB1, S-MFPA1, and S-MFPB1 reduced AST, ALT, and TG levels in mouse serum; increased SOD and GSH-Px activities in the liver; and inhibited MDA production.	[14]
		S-MFPA1	364		50		
		MFPA2	638	Man/Rha/GlcA/GalA/Glc/Xyl/Ara = 8.40:7.00:4.90:11.00:25.30:28.50:14.90	/		
		S-MFPA2	731		/		
		MFPB1	165	Man/Rha/GalA/Glc/Xyl = 9.90:3.60:6.60:43.30:36.50	50		
		S-MFPB1	272		50		
		MFPB2	380	Man/Rha/GlcA/GalA/Glc/Xyl/Ara = 2.80:9.50:5.70:6.10:7.40:45.50:23.00	/		
		S-MFPB2	458		/		
Orchidaceae	*Dendrobium officinale*	DOP	245	D-Glcp/D-Manp = 1.00:4.18	400	Maintained the relative balance between ROS and antioxidants and improved GSH level in the liver.	[18]
Orchidaceae	*Bletilla striata*	BSPS	/	/	125, 250, 500	Reduced oxidative damage and lipid deposition in the liver. High-dose group had the best effect, with increased SOD and GSH activities and decreased ALT and TG levels.	[19]
Asphodelaceae	*Aloe barbadensis*	AP1 AP2 AP3	/	/	12.5, 25, 50	Prolonged drunkenness incubation period shortened sleeping time and waking time of mice.	[20]
Menispermaceae	*Sinomenium acutum*	*Sinomenium acutum* polysaccharides	/	/	100, 200, 400	Improved cell degeneration and necrosis degree of mice in medium-dose and high-dose groups.	[21]
Schisandraceae	*Schisandra chinensis*	SCAP-2	/	/	5, 10, 20	High-dose SCAP-2 decreased ALT and AST activities in serum, increased SOD activity, reduced MDA content in serum and the liver, and reduced TG content in the liver.	[22]
Schisandraceae and Fabaceae	*Schisandra chinensis* and *Radix Astragali*	ASP	/	/	100	Reduced liver index and ALT and AST levels in serum, elevated GSH content, reduced TG and MDA levels in the liver, and improved pathological changes in liver tissues.	[23]
Apiaceae	*Angelica sinensis*	ASP	/	/	200	Reduced serum ALT, AST, and TG levels; prevented hepatic steatosis and fat accumulation; and up-regulated hepatic HIF-1α protein expression.	[15]
Apiaceae	*Angelica sinensis*	Cur/ACNPs (Cur-loaded amphiphilic cholesteryl hemisuccinate-Angelica sinensis polysaccharide self-assembled nanoparticles)	/	/	1.84	Reduced AST, ALT, and MDA levels; increased GSH content; and increased GSH-Px and SOD activities.	[24]
Apocynaceae	*Cynanchum bungei*	*Cynanchum bungei* polysaccharides	/	/	100, 250, 500	Medium-dose and high-dose polysaccharides increased ALT and AST activities.	[25]
Araliaceae and Rhamnaceae	*Panax quinquefolius* and *Hovenia dulcis*	AGH	/	/	100	Increased serum AST and ALT levels; reduced serum TG and liver MDA levels; increased SOD and GSH-Px activities; enhanced liver GSH level; and inhibited expression of p-ERK, p-JNK, and p-P38.	[26]
Polyporaceae	*Ganoderma lucidum*	*Ganoderma lucidum* polysaccharides	/	/	150	Decreased levels of serum AST, ALT, TG, TC, free fatty acids, IL-1β, IL-6, and TNF-α; inhibited NLRP3 inflammatory corpuscle protein expression in the liver; and alleviated inflammatory response and liver fat deposits.	[16]
Polyporaceae	*Ganoderma lucidum*	GLFPS	/	/	200	Inhibited levels of ALT, AST, TG, TC, and ADH in serum and prevented alcoholic liver injury by regulating metabolism of choline, glycerophospholipids, and some ABC transporters.	[17]

Note: Man, mannose; Rha, rhamnose; Glc, glucose; Xyl, xylose; GlcA, glucuronic acid; GalA, galacturonic acid; Glc, glucose; Ara, arabinose; D-Glcp, D-glucose; D-Manp, D-mannose; p-ERK, phosphorylated extracellular regulated protein kinases; p-JNK, phosphorylated c-JUN N-terminal kinase; p-P38, phosphorylated P38 protein kinases; /, no data; *Ganoderma lucidum* is a fungus (not a plant).

**Table 2 ijms-24-16530-t002:** PEMPs with hepatoprotective activities against subacute alcoholic liver injury.

Family	Source of Polysaccharides (Species)	Names of Polysaccharides	Molecular Weight (kDa)	Monosaccharide Composition	Dose (mg/kg)	Bioactivity	Reference
Araliaceae	*Panax notoginseng*	PNPS-0.5 M	2600	Rha/Ara/Xly/Man/Gal = 12.30:33.80:25.80:5.60:22.50	100	Reduced ALT, AST, TG, and MDA levels; elevated the activities of SOD and GSH-Px; and activated Nrf2 signaling as a protective mechanism against Cyp2e1 toxicity.	[28]
Orchidaceae	*Dendrobium huoshanense*	DHP	/	Glc/Man/Gal/Ara = 1.00:0.07:0.11:0.03	50, 100, 200	Among all the *Dendrobium* polysaccharides, DHP and DOP possessed the highest potential for protecting the liver from hepatotoxicity caused by alcohol intake, which was evidenced as follows: (a) decreased levels of ALT, AST, ALP, TBIL, TC, TG, and LDL-C in serum, as well as TC, TG, MDA, CYP2E1, TNF-α, and IL-1β in hepatic tissues; (b) increased levels of HDL-C in serum and SOD, CAT, GSH-PX, GR, GST, GSH, ADH, and ALDH in hepatic tissues; and (c) ameliorated histopathological changes in hepatic tissues.	[29]
	*Dendrobium officinale*	DOP		Glc/Man/Gal/Ara = 1.00:0.13:0.12:0.03			
	*Dendrobium fimbriatum*	DFP		Glc/Man/Gal/Ara = 1.00:0.29:0.02:0.01			
	*Dendrobium chrysotoxum*	DCP		Glc/Man/Gal = 1.00:0.42:0.04			
	*Dendrobium nobile*	DNP		Glc/Man/Gal/Ara/Rha/Xyl = 1.00:0.14:0.10:0.07:0.06:0.04			
	*Dendrobium moniliforme*	DMP		Glc/Man/Gal = 1.00:2.25:0.08			
Orchidaceae	*Dendrobium huoshanense*	DHP	/	/	11, 21, 64	Suppressed the changes in ALT, AST, ALP, HDL-C, LDL-C, TC, and TG levels in serum; enhanced the activities of ADH, ALDH, SOD, and GSH-Px in the liver; and inhibited the decrease in GSH level and the increase in MDA level in the liver.	[30]
Orchidaceae	*Dendrobium huoshanense*	DHP	/	/	400	Decreased the ratio of liver weight to body weight; reduced the levels of serum AST, TG, TBIL, and LDL; alleviated hepatic steatosis; and regulated the expression of Cbs and Ldhd genes.	[31]
Orchidaceae	*Dendrobium huoshanense*	DHP	22	Glc/Man/Gal = 2.40:1.00:1.00	400	Alleviated early steatosis and inflammation in liver histology and ameliorated the altered metabolic levels, particularly those involving phosphocholine and L-proline.	[32]
Orchidaceae	*Gastrodia elata*	GEP-2	/	/	25, 50, 100	Reduced the levels of ALT and AST in serum and the content of TG, decreased the level of MDA, and increased the activity of SOD.	[33]
Phallaceae	*Dictyophora rubrovalvata*	DRP	/	/	100, 200, 400	Decreased the levels of AST, ALT, and TG; increased the levels of SOD and GSH; decreased the contents of MDA, TNF-α, and IL-6; and improved the pathological phenomena of liver cellular degeneration and necrosis.	[34]
Rosaceae	*Rosa rugosa*	*Rosa rugosa* polysaccharides	/	/	100, 300, 500	Increased the activities of SOD, GSH, and GSH-Px; decreased the contents of NO and MDA; and decreased the serum levels of IL-6, IL-1β, and TNF-α.	[35]
Polyporaceae	*Poria cocos*	PCP	/	/	50, 200	Reduced ALT, AST, MDA, IL-6, and TNF-α; potentiated activity of SOD; regulated the expression of CYP2E1; and inhibited TLR4/NF-κB inflammatory signaling pathway.	[27]
Omphalotaceae	*Lentinula edodes*	lentinan	/	/	750, 1500, 3000	Decreased MDA and TG, increased GSH content, regulated liver metabolism, and alleviated hepatocyte apoptosis.	[36]
Ciavieps purpurea	*Cordyceps militaris*	CMSP	/	/	150, 300, 600	Reduced the pathological damage and inflammatory response of liver tissue, and the pathological improvement was positively correlated with the dose.	[37]

Note: Glc, glucose; Man, mannose; Gal, galactose; Ara, arabinose; Rha, rhamnose; Xyl, xylose; ALP, alkaline phosphatase; HDL-C, high-density lipoprotein cholesterol; LDL-C, low-density lipoprotein cholesterol; LDL, low-density lipoprotein; Cbs, cystathionine beta-synthase; Ldhd, D-lactate dehydrogenase; Nrf2, nuclear factor E2-related factor 2; /, no data; *Poria cocos*, *Lentinula edodes,* and *Cordyceps militaris* are fungi (not plants).

**Table 3 ijms-24-16530-t003:** PEMPs with hepatoprotective activities against subchronic alcoholic liver injury.

Family	Source of Polysaccharides (Species)	Names of Polysaccharides	Molecular Weight (kDa)	Monosaccharide Composition	Dose (mg/kg)	Bioactivity	Reference
Rhamnaceae	*Ziziphus jujube*	Jujube polysaccharides	/	/	4000, 8000, 16,000	Increased activities of SOD, CAT, and GSH-Px, but decreased MDA content.	[38]
Liliaceae	*Allium sativum*	Garlic polysaccharides	10	Fru/Gal/GalA = 307.00:25.00:32.00	150, 250	Reduced MDA, TC, TG, and LDL levels; increased SOD, GSH-Px, and GSH levels; and inhibited TGF-b1 and TNF-α expression.	[39]
Solanaceae	*Lycium barbarum*	LBPs	/	/	300	Elevated GSH, SOD, CAT, and GSH-Px levels, but reduced MDA level.	[40]
Asteraceae	*Arctium lappa*	*Arctium lappa* polysaccharides	/	/	100, 300, 900	Antagonized alcohol-induced hepatic steatosis and reduced TG content in liver tissues.	[42]
Acanthaceae	*Dicliptera chinensis*	*Dicliptera chinensis* polysaccharides	/	/	100, 200, 300	Reduced serum ALT, AST, and MDA levels; elevated GSH-Px and SOD activities; decreased liver TNF-α and IL-6 levels; and alleviated pathological lesions in the liver.	[43]

Note: Fru, fructose; Gal, galactose; GalA, galacturonic acid; LDL, low-density lipoprotein; TGF-b1, transforming growth factor-beta 1; /, no data.

**Table 4 ijms-24-16530-t004:** PEMPs with hepatoprotective activities against chronic alcoholic liver injury.

Family	Source of Polysaccharides (Species)	Names of Polysaccharides	Molecular Weight (kDa)	Monosaccharide Composition	Dose (mg/kg)	Bioactivity	Reference
Asphodelaceae	*Aloe vera*	AVGP	/	/	10, 30	Attenuated levels of serum aminotransferases and TG; ameliorated histopathological alterations; up-regulated hepatic expression of lipolytic genes (AMPK-α2 and PPAR-α); decreased MDA level; increased GSH and SOD levels; and mitigated alcohol-induced inflammation via reduction in LPS and TNF-α, down-regulation of TLR-4 and MyD88, and up-regulation of IκB-α.	[44]
Solanaceae	*Lycium barbarum*	LBPs	/	/	250, 500	Decreased levels of TG, TC, TNF-α, MDA, ALT, AST, and CYP2E1; increased levels of SOD, CAT, GSH-PX, GSH, and ADH; and alleviated liver tissue lesions.	[46]
Liliaceae	*Allium sativum*	garlic polysaccharides	/	/	100, 150, 200	Decreased liver index and serum ALT and AST activities, increased GSH and GSH-Px levels, decreased MDA level, and alleviated pathological liver alterations at all doses. Increased body weight and Na^+^, K^+^-ATPase activity at high and medium doses.	[47]
Lauraceae	*Cinnamomum camphora*	*camphora* leaf polysaccharides	/	/	25.0, 37.5, 50.0	Decreased levels of IL-1β, IL-6, TNF-α, TG, ALT, and AST; increased GSH level; and improved SIRT1 expression and AMPK-alpha phosphorylation.	[48]
Lamiaceae	*Perilla frutescens*	*Perilla frutescens* leaf polysaccharides	/	/	300, 600	Reduced serum ALT, AST, TG, TC, and LDL levels; increased HDL content; decreased IL-1β, IL-6, TNF-α, and MDA contents in the liver; improved SOD and GSH-Px activities; facilitated expression of p-AMPKα/AMPKα and SIRT1 in the liver; and inhibited expression of SREBP1c.	[49]
Polyporaceae	*Ganoderma lucidum*	SGPs (mycelium polysaccharides)	/	/	125, 250, 500	Increased activities of CAT, SOD, and GSH-PX in the liver and serum at medium and high doses; decreased activities of ALT and AST; and reduced contents of MDA, TG, TG, IL-6, IL-1β, and TNF-α at high dose.	[45]

Note: ATP, adenosine triphosphate; AMPK, adenosine monophosphate-activated protein kinase; PPAR-α, peroxisome proliferator-activated receptor-alpha; IκB-α, I kappa B-alpha; SIRT1, silent information regulator sirtuin 1; LDL, low-density lipoprotein; HDL-C, high-density lipoprotein; SREBP1c, sterol regulatory element-binding protein-1c; /, no data; *Ganoderma lucidum* is a fungus (not a plant).

## Data Availability

The data presented in this study are available on request from the corresponding author.

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
