# Peer review of "Protective Effects and Mechanism of Polysaccharides from Edible Medicinal Plants in Alcoholic Liver Injury: A Review"

_ijms, 2023, doi:10.3390/ijms242216530_

Round 1
Reviewer 1 Report
Comments and Suggestions for Authors
Reviewer comments and suggestions
The authors in this review discussed the protective effects of polysaccharides from edible medicinal plants (PEMPs) on acute, subacute, subchronic and chronic acute liver injury (ALI). Structure-activity relationship (SAR), as well as the safety of PEMPs, were discussed. Moreover, they also included the mechanism underlying the hepatoprotective activity of polysaccharides from edible medicinal plants and strategies for further research into practical application were also briefly explained.
Overall, the manuscript was well written. However, a few major concerns/comments needed to be explained/modified.
- Line 47-48 The authors need to briefly explain these points
- Section 3.2.2 Attenuation of oxidative stress caused by alcohol intake is this correct, please check it
- In lines 247-248 the author mentions a few studies, but they did not include any references or explanations
- Section 4.2 The section should be well described.
- A few references should be modified such as (why they are underlined 7, 11,62,63),30, and others
- Comments for Figure 1 Please add the sources of the figures such as reference
Author Response
Responses to Reviewer 1
Journal: International Journal of Molecular Sciences
Special Issue: Current Research in Pharmacognosy: A Focus on Biological Activities
Manuscript Title: Protective effects and mechanism of polysaccharides from edible medicinal plants on alcoholic liver injury: A review
Manuscript No: ijms-2665706
Dear reviewers,
First of all, we are grateful to you for your kind work. We have revised our manuscript according to the editor and the reviewers. The comments and our answer are listed as follows:
Comment No. |
Reviewers’ comments |
Authors’ responses |
Reviewer 1 |
||
1 |
The authors in this review discussed the protective effects of polysaccharides from edible medicinal plants (PEMPs) on acute, subacute, subchronic and chronic acute liver injury (ALI). Structure-activity relationship (SAR), as well as the safety of PEMPs, were discussed. Moreover, they also included the mechanism underlying the hepatoprotective activity of polysaccharides from edible medicinal plants and strategies for further research into practical application were also briefly explained. Overall, the manuscript was well written. However, a few major concerns/comments needed to be explained/ modified. Line 47-48 The authors need to briefly explain these points. |
We are grateful to the reviewer 1 for his/her valuable comments. According to the reviewer, we have revised our manuscript. |
2 |
Section 3.2.2 Attenuation of oxidative stress caused by alcohol intake is this correct, please check it. |
The sentence has been modified. |
3 |
In lines 247-248 the author mentions a few studies, but they did not include any references or explanations. |
The sentences have been explained as suggested. |
4 |
Section 4.2 The section should be well described. |
The section has been rephrased according to the reviewer. |
5 |
A few references should be modified such as (why they are underlined 7, 11, 62, 63) and others. |
The references (7, 11, 52, 56, 57, 62, 63) in our manuscript have been refreshed as suggested. |
6 |
Comments for Figure 1 Please add the sources of the figures such as reference. |
Figure 1 is draw by us, so there is no reference. |
In the revised version of our manuscript, the revisions have been highlighted with red font throughout the manuscript. If you have any queries about our manuscript, please do not hesitate to contact me.
Thank you again for your kind work.
Best regards!
Sincerely yours,
Hua-Feng Zhang (Director, PI, Chair Professor, Ph.D.)
Director and principal investigator (PI) of International Joint Research Center of Shaanxi Province for Food and Health Sciences;
Chair professor and chief expert of Provincial Expert Workstation for Hua-Feng Zhang, Academician and Expert Workstation in Puer City of Yunnan Province;
Shaanxi Normal University, Xi’an City, Shaanxi Province, P.R. China
Website: http://ijrc.snnu.edu.cn/info/1014/1066.htm
E-mail: isaacsau@sohu.com; ijrc@snnu.edu.cn

Reviewer 2 Report
Comments and Suggestions for Authors
In this study, Su et al. thoroughly review the protective impacts of polysaccharides derived from edible medicinal plants (PEMPs) on alcoholic liver injury (ALI). The authors explore the protective effects of various PEMPs on acute, subacute, subchronic, and chronic ALI, making this a valuable resource for researchers in this area. The review also delves into the pathogenesis of ALI and the structure-activity relationship (SAR) of PEMPs, providing insights into the mechanisms underlying their hepatoprotective properties. Moreover, the review highlights the safety of PEMPs, emphasizing the need for systematic safety evaluations for their potential use in treating ALI. However, several areas require improvement:
- The abstract should include more specific information about the main findings or conclusions of the review to enhance reader understanding.
- The introduction should clearly state the purpose or objectives of the review to define its scope and focus.
- The tables should provide more detailed footnotes or legends to help readers better understand the presented information.
- The conclusion should offer more specific recommendations for future research or practical applications of the findings.
- The manuscript requires improvements in language and grammar. The typographical errors and inconsistencies need correction. For instance, the term "PEMPs" is occasionally written as "PEMPS". Additionally, some sentences are overly long and could be broken down into smaller sentences for clarity and improved readability.
- The manuscript requires improvements in language and grammar. The typographical errors and inconsistencies need correction. For instance, the term "PEMPs" is occasionally written as "PEMPS". Additionally, some sentences are overly long and could be broken down into smaller sentences for clarity and improved readability.
Author Response
Responses to Reviewers
Journal: International Journal of Molecular Sciences
Special Issue: Current Research in Pharmacognosy: A Focus on Biological Activities
Manuscript Title: Protective effects and mechanism of polysaccharides from edible medicinal plants on alcoholic liver injury: A review
Manuscript No: ijms-2665706
Dear reviewers,
First of all, we are grateful to you for your kind work. We have revised our manuscript according to the editor and the reviewers. The comments and our answer are listed as follows:
Comment No. |
Reviewers’ comments |
Authors’ responses |
Reviewer 2 |
||
1 |
In this study, Su et al. thoroughly review the protective impacts of polysaccharides derived from edible medicinal plants (PEMPs) on alcoholic liver injury (ALI). The authors explore the protective effects of various PEMPs on acute, subacute, subchronic, and chronic ALI, making this a valuable resource for researchers in this area. The review also delves into the pathogenesis of ALI and the structure-activity relationship (SAR) of PEMPs, providing insights into the mechanisms underlying their hepatoprotective properties. Moreover, the review highlights the safety of PEMPs, emphasizing the need for systematic safety evaluations for their potential use in treating ALI. However, several areas require improvement. The abstract should include more specific information about the main findings or conclusions of the review to enhance reader understanding. |
We are grateful to the reviewer 2 for his/her valuable comments. According to the reviewer, the abstract has been modified as suggested. |
2 |
The introduction should clearly state the purpose or objectives of the review to define its scope and focus. |
According to the reviewer, we have supplemented Introduction with some sentences in order to define its scope and focus. |
3 |
The tables should provide more detailed footnotes or legends to help readers better understand the presented information. |
We have added some detailed footnotes according to the reviewer. |
4 |
The conclusion should offer more specific recommendations for future research or practical applications of the findings. |
According to the reviewer, we have rephrased the Conclusion. |
5 |
The manuscript requires improvements in language and grammar. |
In order to improve the quality of our manuscript (including the English language), we have invited Dr. Xiao Ying to revise the manuscript. She is a professor in Macau University of Science and Technology. |
6 |
The typographical errors and inconsistencies need correction. For instance, the term "PEMPs" is occasionally written as "PEMPS". |
We have checked and modified the typographical errors and inconsistencies according to the reviewer. |
7 |
Additionally, some sentences are overly long and could be broken down into smaller sentences for clarity and improved readability. |
According to the reviewer, we have modified the sentences. |
In the revised version of our manuscript, the revisions have been highlighted with red font throughout the manuscript. If you have any queries about our manuscript, please do not hesitate to contact me.
Thank you again for your kind work.
Best regards!
Sincerely yours,
Hua-Feng Zhang (Director, PI, Chair Professor, Ph.D.)
Director and principal investigator (PI) of International Joint Research Center of Shaanxi Province for Food and Health Sciences;
Chair professor and chief expert of Provincial Expert Workstation for Hua-Feng Zhang, Academician and Expert Workstation in Puer City of Yunnan Province;
Shaanxi Normal University, Xi’an City, Shaanxi Province, P.R. China
Website: http://ijrc.snnu.edu.cn/info/1014/1066.htm
E-mail: isaacsau@sohu.com; ijrc@snnu.edu.cn
